# Improved Solutions to the Linearized Boussinesq Equation with Temporally Varied Rainfall Recharge for a Sloping Aquifer

**Ming-Chang Wu and Ping-Cheng Hsieh ***

Department of Soil and Water Conservation, National Chung Hsing University, Taichung 40227, Taiwan; ry2312611@gmail.com
* Correspondence: pchsieh@nchu.edu.tw; Tel.: +886-4-22840381-107

**Abstract:** Sloping unconfined aquifers are commonly seen and well investigated in the literature. In this study, we propose a generalized integral transformation method to solve the linearized Boussinesq equation that governs the groundwater level in a sloping unconfined aquifer with an impermeable bottom. The groundwater level responses of this unconfined aquifer under temporally uniform recharge or nonuniform recharge events are discussed. After comparing with a numerical solution to the nonlinear Boussinesq equation, the proposed solution appears better than that proposed in a previous study. Besides, we found that the proposed solutions reached the convergence criterion much faster than the Laplace transform solution did. Moreover, the application of the proposed solution to temporally changing rainfall recharge is also proposed to improve on the previous quasi-steady state treatment of an unsteady recharge rate.

**Keywords:** groundwater; rainfall recharge; unconfined aquifer; generalized integral transformation method (GITM)

## 1. Introduction

Groundwater level has been widely investigated by experimental or field data collection, numerical methods, and analytical approaches. In general, the groundwater level is difficult to estimate and predict compared with the water on the ground. Various items of groundwater level estimation equipment are highly expensive; thus, substantial financial support is required. Therefore, some researchers study groundwater problems mainly by employing numerical methods but others prefer to apply analytical approaches.

It is necessary to quantify the hydrological processes under the hillside and develop appropriate approaches to describe these processes. Regarding this subject, many models have been developed over the past 40 years. Paniconi and Wood [1] developed a three-dimensional finite element numerical model based on the Richards equation to deal with catchment scale simulations. Such a large numerical code consumes much computer time and memory, and it is very hard to examine the validation of the code. Brutsaert [2] derived an analytical solution to the linearized Boussinesq equation and studied the response of the groundwater flow per unit width of the slope with consideration of zero water depth at the downstream boundary condition, corresponding to the free drainage of the unconfined aquifer. The analytical method provides a powerful framework for analyzing the effects of different features of the slope on its hydrological response shape.

Chapman [3] used a simple empirical method to build a power–law relationship between storage and discharge in a hillside. Later on, Berne et al. [4] converted the linearized Boussinesq equation into a hillslope-storage Boussinesq equation and presented the moments of the characteristic response

function (CRF) to study the latter equation with fixed recharge, following the research of Troch et al. [5]. In fact, the thickness of the free seepage surface is known to vary with the configuration of the hillside by referring to Chapman [3]. Recently, Dralle et al. [6] derived a new analytical solution to the linearized hillslope Boussinesq equation with spatially variable recharge by the method of eigenfunction expansion, and discussed the hydrologic response of topography to base flow discharge properties. In their study, they claimed that their solutions exactly reproduce previous results, e.g., Verhoest and Troch [7] and Troch et al. [5], for the case of spatially uniform recharge, and perfectly match the numerical solutions by a finite difference scheme for the case of spatially variable recharge. However, the linearization constant, $\varepsilon = 2/3$ in Verhoest and Troch [7] but $\varepsilon = 1$ in Dralle et al. [6], is different, and in their modeling scenarios they hypothesized the spatially and temporally variable recharge and simplified its distribution in two intervals only.

Because the groundwater level is mainly influenced by flow seepage and external recharge, some researchers recently considered rainfall recharge. Vehoest and Troch [7] performed the Laplace transformation to solve the linearized Boussinesq equation to estimate the groundwater level under the effect of rainfall recharge. In their study, Arfken and Weber [8] applied a complex inversion formula to obtain the inverse Laplace transform by using a Bromwich integral. Moreover, the transient groundwater level was approximated by a steady state condition that generated the same outflow. Zissis et al. [9] discussed a groundwater table that was affected by a river's constant recharge and the variation in the water level of that river. They also linearized the nonlinear Boussinesq equation. The results proved that under the same conditions and when not considering rainfall recharge, the solution of the linearized equation is very similar to that of the nonlinear equation. However, the discrepancy of these two solutions becomes apparent for a case that has a high amount of rainfall recharge and a mild slope. Bansal and Das [10] proposed a groundwater model to discuss the water table in an unconfined sloping aquifer under constant recharge and seepage from a stream in which the water level varied. In their model, the linearized Boussinesq equation was also employed as a governing equation.

Most published studies on groundwater problems have focused on uniform recharge, but this is not sufficient to delineate various real conditions such as rainfall recharge. Kazezyilmaz-Alhan [11] used the Heaviside function (also known as the unit step function) to represent temporally changing rainfall events, and the transient variation in the overland flow was discussed by employing the diffusion wave theory. Such techniques are considered to treat the source term in this study.

On the basis of Chapman's study [12], when the angle of the impermeable bottom slope is less than 30°, the flow in the aquifer appropriately conforms to Dupuit's assumptions. Therefore, a modified one-dimensional Boussinesq equation is presented for groundwater flow in a sloping aquifer. In this text, the first section explains the research background, motivation and purpose, content and basic structure of the paper. The second section describes the mathematical derivation of the presented problem and its analytical solution as well as the introduction of the general integral transformation method (GITM). In the third section, the differences among the present analytical solution, the previous analytical solution and the nonlinear numerical solution are discussed, and the groundwater level and flow fluctuations under different conditions are simulated. Finally, the results of this research are concluded.

## 2. Mathematical Formulation

### 2.1. Conceptual and Mathematical Models

The groundwater flow in a sloping unconfined aquifer (Figure 1) based on Darcy's law is governed by (see Childs [13])

$$q = -KH_w\left(cos\theta\frac{\partial H_w}{\partial x} + sin\theta\right) \tag{1}$$

and the flow satisfies the following continuity equation according to the law of mass conservation:

$$n\frac{\partial H_w}{\partial t} + \frac{\partial q}{\partial x} = r \tag{2}$$

where $n$ is the drainable or effective porosity (-), $q$ is the flow rate in the $x$ direction per unit width of the aquifer (L$^2$/T), $K$ is the hydraulic conductivity (L/T), $H_w$ is the elevation of the groundwater table measured perpendicularly to the underlying impermeable layer (L), $\theta$ is the inclined angle of the aquifer bottom (-), and $r = r(t)$ is the rainfall recharge rate (L/T).

To investigate the groundwater flow problem in a sloping unconfined aquifer, we substituted Equation (1) into Equation (2) and obtained a Boussinesq equation for a sloping aquifer by assuming no spatial variability in $K$, $n$, and $\theta$:

$$\frac{\partial H_w}{\partial t} = \frac{K}{n}\left[cos\theta\frac{\partial}{\partial x}\left(H_w\frac{\partial H_w}{\partial x}\right) + sin\theta\frac{\partial H_w}{\partial x}\right] + \frac{r}{n} \tag{3}$$

Brutsaert [2] stated that the nonlinear term $H_w\partial H_w/\partial x$ on the right-hand side of Equation (3) can be linearized by changing the first $H_w$ to $\varepsilon D$. $D$ is the thickness of the initially saturated aquifer, and $\varepsilon$ is a linearization constant given by $0 < \varepsilon < 1$. Thus, Equation (3) can be given as follows:

$$\frac{\partial H_w}{\partial t} = \frac{K}{n}\left(\varepsilon Dcos\theta\frac{\partial^2 H_w}{\partial x^2} + sin\theta\frac{\partial H_w}{\partial x}\right) + \frac{r}{n} \tag{4}$$

Verhoest and Troch [7] improved on the study by Brutsaert [2] by adding a constant recharge to the aquifer. In their study, they assumed that water initially filled a rectangular aquifer to a depth of $D - h$, as displayed in Figure 1. Moreover, $h$ was the distance from the ground surface to the average groundwater level. Moreover, they assumed that a sudden drawdown at the outlet ($x = 0$) of the aquifer caused the depth of the water level to be zero, and a zero-inflow boundary existed at the hilltop ($x = L$). Hence, the initial condition was

$$H_w = D - h, 0 < x < L, \ t = 0 \tag{5}$$

and the boundary conditions were

$$H_w = 0, \ x = 0, \ t > 0 \tag{6}$$

$$q = 0, \ x = L, \ t > 0 \tag{7}$$

In our present study, we utilized the Heaviside function $u(t)$ to represent the temporally changing rainfall recharge rate:

$$r(t) = \sum_{i=1}^{N} r_i[u(t - t_{i-1}) - u(t - t_i)] \tag{8}$$

Moreover, by substituting $\alpha = K\varepsilon Dcos\theta/n$ and $U = Ksin\theta/n$ in Equation (4), we obtained the following:

$$\frac{\partial H_w}{\partial t} = \alpha\frac{\partial^2 H_w}{\partial x^2} + U\frac{\partial H_w}{\partial x} + \frac{1}{n}\sum_{i=1}^{N} r_i[u(t - t_{i-1}) - u(t - t_i)] \tag{9}$$

To eliminate the first order derivative of $x$, we set

$$H_w(x, t) = e^{\frac{-U}{2\alpha}x}e^{\frac{-U^2}{4\alpha}t}H_v(x, t) \tag{10}$$

By substituting Equation (10) into Equation (9), the initial condition provided by Equation (5), and the boundary conditions given in Equations (6) and (7), we obtained the following:

$$\frac{\partial^2 H_v(x,t)}{\partial x^2} + \frac{\sum_{i=1}^{N} r_i[u(t-t_{i-1}) - u(t-t_i)]}{\alpha n} e^{\frac{U}{2\alpha}x} e^{\frac{U^2}{4\alpha}t} = \frac{1}{\alpha}\frac{\partial H_v(x,t)}{\partial t} \tag{11}$$

$$H_v(x,0) = e^{\frac{U}{2\alpha}x}(D-h),\ 0 < x < L \tag{12}$$

$$H_v(0,t) = 0,\ t > 0 \tag{13}$$

$$2\alpha\frac{\partial H_v(L,\ t)}{\partial x} + UH_v(L,t) = 0, \tag{14}$$

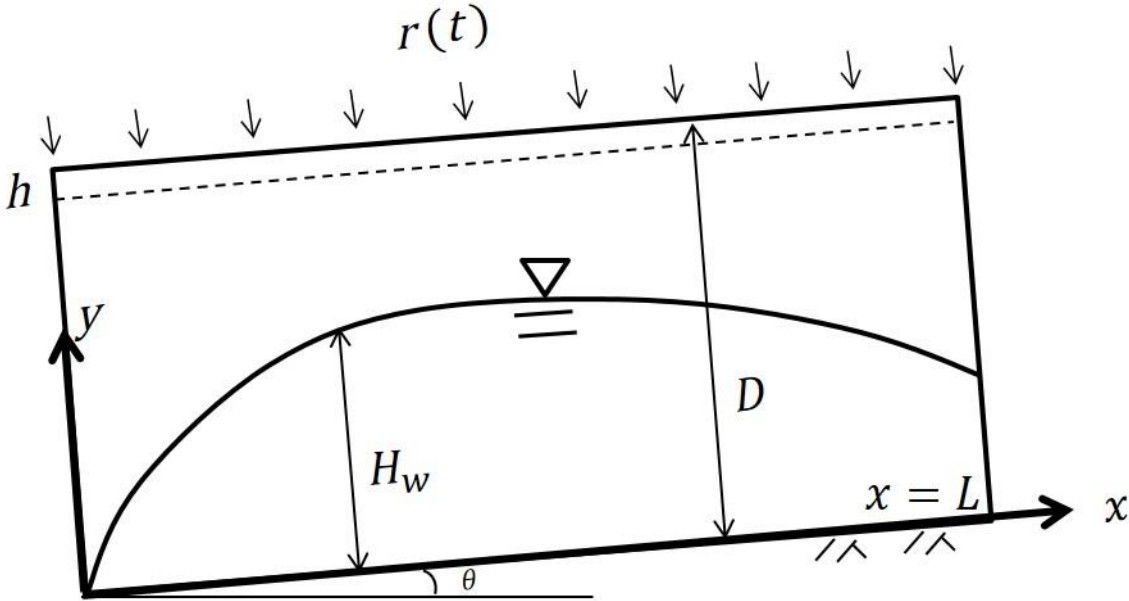

**Figure 1.** Schematic diagram of a sloping aquifer.

## 2.2. Present Improved Solutions

In the present study, we employed the generalized integral transformation (GITM) of Özisik [14] to solve Equation (11) in terms of the following formulas. The GITM is usually employed to solve boundary value problems of heat conduction, which eliminates the spatially quadratic differential term of the governing equation by inserting a kernel function with a space variable only, and then the partial differential equation is transformed into an ordinary differential equation with a time variable. The ordinary differential equation is of a first-order type and easily solved. In the generalized integral inverse transformation, an infinite series with corresponding eigenvalues is included. In theory, while the infinite series is calculated, its eigenvalues need to be evaluated and summed to a maximum number of terms to obtain a more accurate solution. In fact, the integral-transform technique helps to reach a fast convergence of the infinite series.

Transform formula:

$$\overline{H_v}(\beta_m,t) = \int_{x'=0}^{L} \xi(\beta_m,\ x')H_v(x',\ t)dx' \tag{15}$$

The inverse transform formula can be given as

$$H_v(x,t) = \sum_{m=1}^{\infty} \xi(\beta_m,\ x)\overline{H_v}(\beta_m,\ t) \tag{16}$$

here,

$$\xi(\beta_m,x) \equiv \sqrt{2}\left(\frac{\beta_m^2 + \gamma^2}{L(\beta_m^2 + \gamma^2) + \gamma}\right)^{1/2} \times \sin(\beta_m x) \tag{17}$$

$$\gamma = \frac{U}{2\alpha} \tag{18}$$

where $\beta_m$ is the root of

$$\beta cot(\beta L) = -\gamma \tag{19}$$

for the presented problem. The solution to Equation (11) is

$$H_v(x,t) = \sum_m^{\infty} e^{-\alpha\beta_m^2 t} B_m \eta_m sin(\beta_m x) \left[ (D-h) + \frac{1}{n} \sum_{i=1}^{N} r_i \int_{t_{i-1}}^{t_i} e^{\frac{4\alpha^2\beta_m^2 + U^2}{4\alpha}t'} dt' \right] \tag{20}$$

where

$$B_m = \sqrt{\frac{2(\beta_m^2 + \gamma^2)}{L(\beta_m^2 + \gamma^2) + \gamma}} \tag{21}$$

$$\eta_m = \int_0^L e^{Ux/2\alpha} \times sin(\beta_m x)dx = \frac{2\alpha[2\alpha\beta_m - 2\alpha\beta_m e^{\frac{U}{2\alpha}L}cos(\beta_m L) + Ue^{\frac{U}{2\alpha}L}sin(\beta_m L)]}{U^2 + 4\alpha^2\beta_m^2} \tag{22}$$

By substituting Equation (20) into Equation (10), we obtained

$$H_w(x,t) = e^{\frac{-U}{2\alpha}x} e^{\frac{-U^2}{4\alpha}t} \sum_{m=1}^{\infty} e^{-\alpha\beta_m^2 t} B_m \eta_m sin(\beta_m x) \left[ (D-h) + \frac{1}{n} \sum_i^{N} r_i \int_{t_{i-1}}^{t_i} e^{\frac{4\alpha^2\beta_m^2 + U^2}{4\alpha}t'} dt' \right] \tag{23}$$

After the groundwater level has been estimated, the flow discharge at the outlet can be obtained by integrating Equation (2) as follows.

$$
\begin{aligned}
q &= -L \sum_{i=1}^{N} r_i[u(t - t_{i-1}) - u(t - t_i)] + n \int_0^L \frac{\partial H_w}{\partial t} dx \\
&= -L \sum_{i=1}^{N} r_i[u(t - t_{i-1}) - u(t - t_i)] \\
&\quad - \sum_{m=1}^{\infty} B_m \eta_m \lambda_m \left( \alpha\beta_m^2 + \frac{U^2}{4\alpha} \right) e^{\frac{4\alpha^2\beta_m^2 + U^2}{4\alpha}t} \\
&\quad \times \left[ n(D-h) + \sum_{i=1}^{N} r_i \int_{t_{i-1}}^{t_i} e^{\frac{4\alpha^2\beta_m^2 + U^2}{4\alpha}t'} dt' \right]
\end{aligned}
\tag{24}
$$

with

$$\lambda_m = \int_0^L e^{\frac{-U}{2\alpha}x} \times sin(\beta_m x)dx = \frac{2\alpha[2\alpha\beta_m - 2\alpha\beta_m e^{\frac{-U}{2\alpha}L}cos(\beta_m L) - Ue^{\frac{-U}{2\alpha}L}sin(\beta_m L)]}{U^2 + 4\alpha^2\beta_m^2} \tag{25}$$

The generalized integral transformation method is different from the Laplace transform method and the Fourier transform method, and can directly perform integral operations about the space variable in a finite field, a semi-infinite domain and an infinite domain. In reality, the surface replenishment intensity will change with time; therefore, we used the Heaviside function to represent temporally varying recharge rates as shown in Equations (23) and (24) to analyze groundwater level and flow, respectively.

## 3. Results and Discussions

### 3.1. Comparison of Analytical and Numerical Solutions

To validate the present analytical solutions, we followed the hypothetical case proposed by Verhoest and Troch [7] with $D - h = 1.5$ m, $K = 0.001$ m/s, $n = 0.34$, $r = 3$ mm/h, and $\varepsilon = 2/3$. For comparison, a numerical solution to the nonlinear Boussinesq Equation (3), subjected to the conditions of Equations (5)–(7), was obtained. In the numerical method, we employed the central difference and the upwind scheme in the Swanson and Turke [15] with respect to space. The time

reference was solved by the third-order Total Variation Diminishing Runge-Kutta scheme proposed by Shu and Osher [16].

After the parameters had been substituted into these solutions, the proposed analytical solution better matched the numerical solution than the solution of Verhoest and Troch [7], as depicted in Figure 2, which illustrates the spatial changes of the groundwater levels for various bottom slopes. Figure 2 demonstrates that there is a large discrepancy between the curve of Verhoest and Troch [7] and the curve of the numerical solution for constant recharge. However, the curve of the present solution is closer to the numerical solution. The shift between both solutions is displayed in Figure 2a,b; it decreases as the bottom slope increases. Furthermore, for the case of simulation time of 3 days, the peak value of the present solution is close to that of the numerical solution while the solution of Verhoest and Troch [7] shifts to the right, as indicated in Figure 3. Because they solved the linearized governing equation by the Laplace transform method instead of the fully nonlinear one, their solution obtained a response to the sloping effect slower than the numerical solution. Similar results could be found in Figure 4 for the case of simulation time of 5 days. On the contrary, the present solution by GITM made a response to the sloping effect a little faster than the nonlinear solution owing to the linearization, as shown in Figure 4a.

To quantitatively assess the difference between analytical and numerical solutions, we proceeded to an error analysis by evaluating the relative percentage difference (RPD), which is defined as follows:

$$\text{RPD} = \frac{Hw_{num} - Hw_{ana}}{Hw_{num}} \tag{26}$$

The error analysis of groundwater level between the analytical solutions and the numerical solution is shown in Figures 5–7 for different durations and different slopes. Figure 3 illustrates that the maximum RPD value of the present solution is 12% as $20 < x < 80$ m, but the maximum RPD value of the solution of Verhoest and Troch [7] is 44%. This indicates that the present solution is much better. Comparing Figure 5a,b, we also found that the results of the present solution for $\theta = 6°$ were better than that for $\theta = 2°$. This implies that the accuracy might increase with the slope. A similar tendency could also be found in Figures 6 and 7.

Moreover, while inspecting Figures 2–7 carefully, we found that as the dip angle of the aquifer increases, the difference between Verhoest and Troch [7] and the present solution becomes smaller, and it is speculated that $\varepsilon$ is the key to affect this difference. Verhoest and Troch [7] used a linearization parameter $\varepsilon$ of 2/3 constantly, but Koussis [17] argued that $\varepsilon$ should be affected by the net infiltration, slope, and hydraulic conductivity, and Brutsaert [2] suggested that $H_w$ in the nonlinear term could be replaced by $\varepsilon D$. Such suggestion of linearization will not create too much error in the solutions if the variation of groundwater table is small. Based on the foregoing statement, this study admits $\varepsilon = 0.17$ in the case of $\theta = 2°$ and $\varepsilon = 0.3$ in the case of $\theta = 6°$ to obtain better results.

Figure 8 displays the temporal change of the flow rate at the outlet for various bottom slopes. As can be seen from the figure, the trend of all the three solutions is consistent. However, as the bottom slope increases, the present solution matches the numerical solution much better than the solution of Verhoest and Troch [7]. Although there is still a little discrepancy between the analytical linearized solution and the numerical nonlinear solution, the presented solution improves the analytical results of the previous study. To sum up, the present solution is closer to the numerical solution of the nonlinear Boussinesq equation within a short period and tends to be constant and overlap with the numerical results for a long time. Therefore, the present solutions seem to be more feasible.

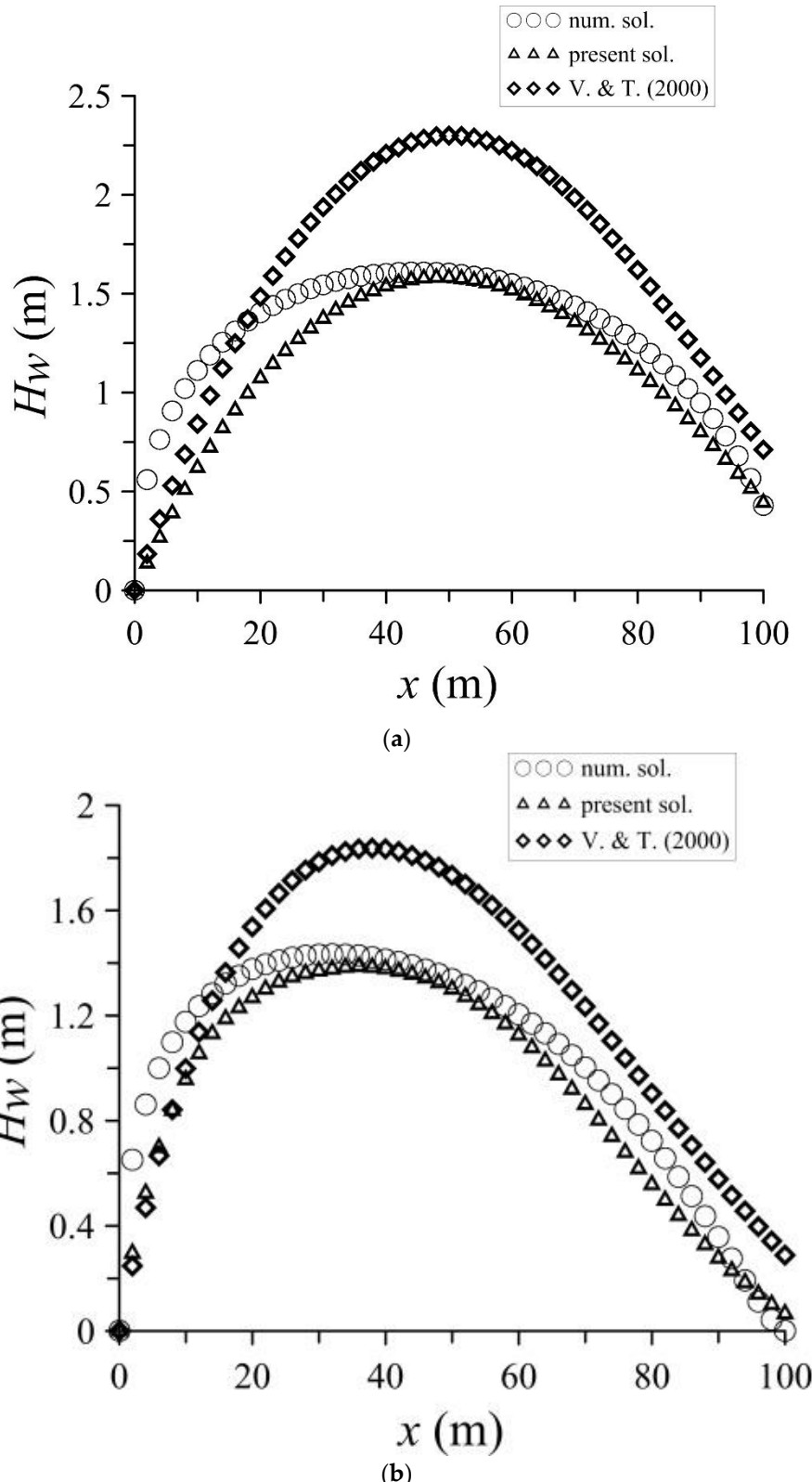

**Figure 2.** Spatial variation in the groundwater table under constant recharge for (**a**) $\theta = 2°$ (**b**) $\theta = 6°$ ($t = 1$ day).

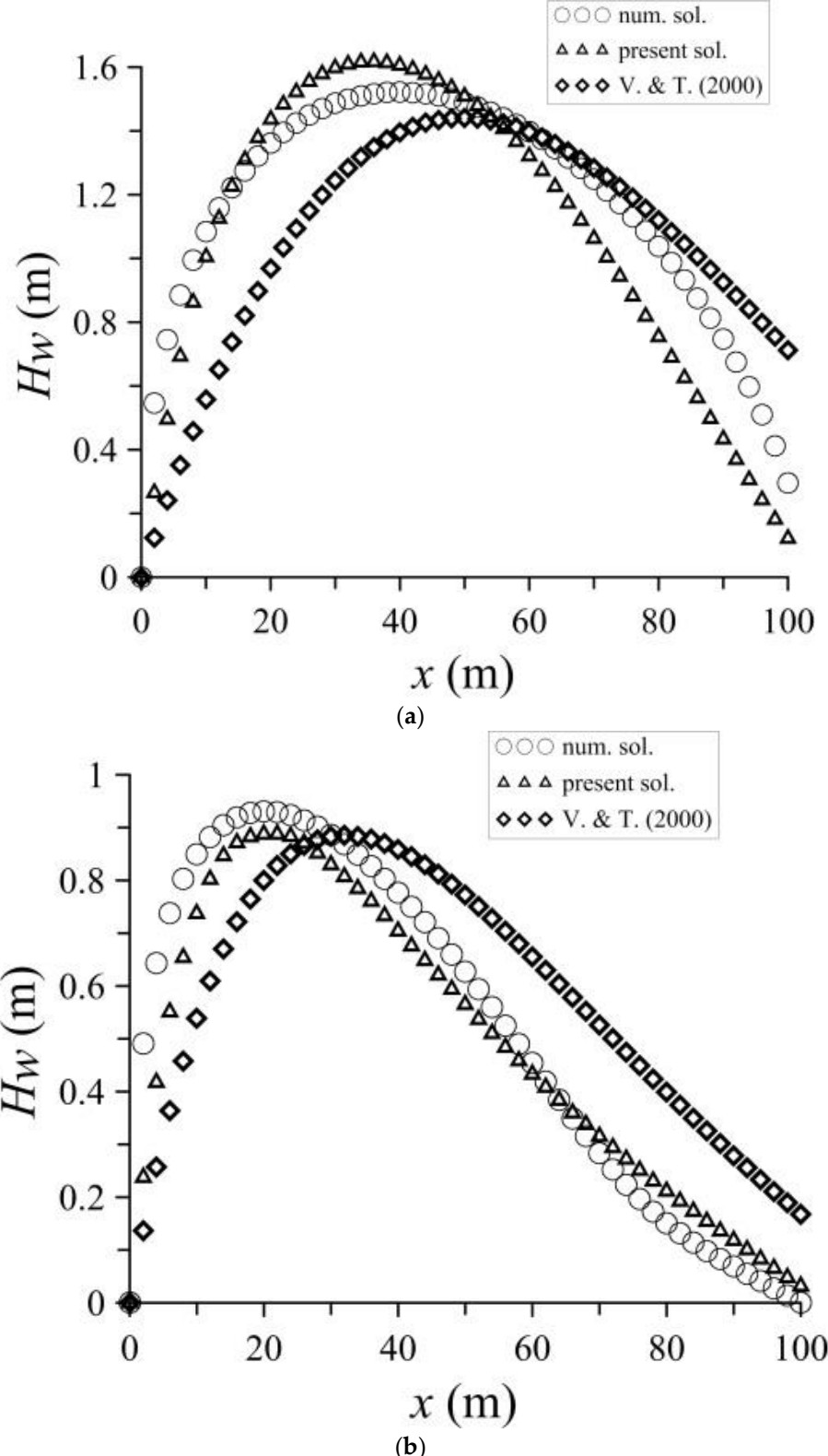

**Figure 3.** Spatial variation in the groundwater table under constant recharge for (**a**) $\theta = 2°$ (**b**) $\theta = 6°$ ($t = 3$ days).

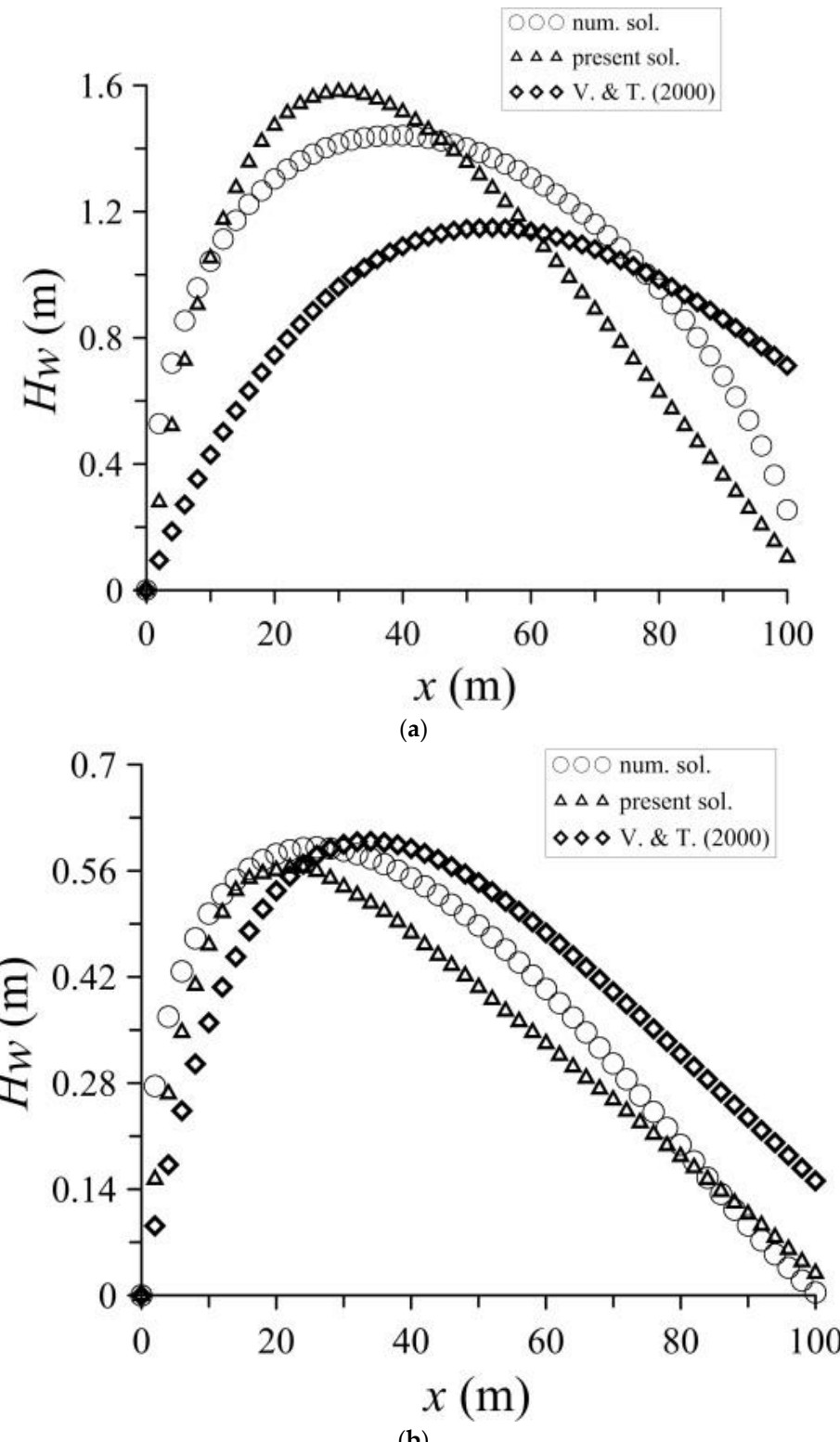

**Figure 4.** Spatial variation in the groundwater table under constant recharge for (**a**) $\theta = 2°$ (**b**) $\theta = 6°$ ($t = 5$ days).

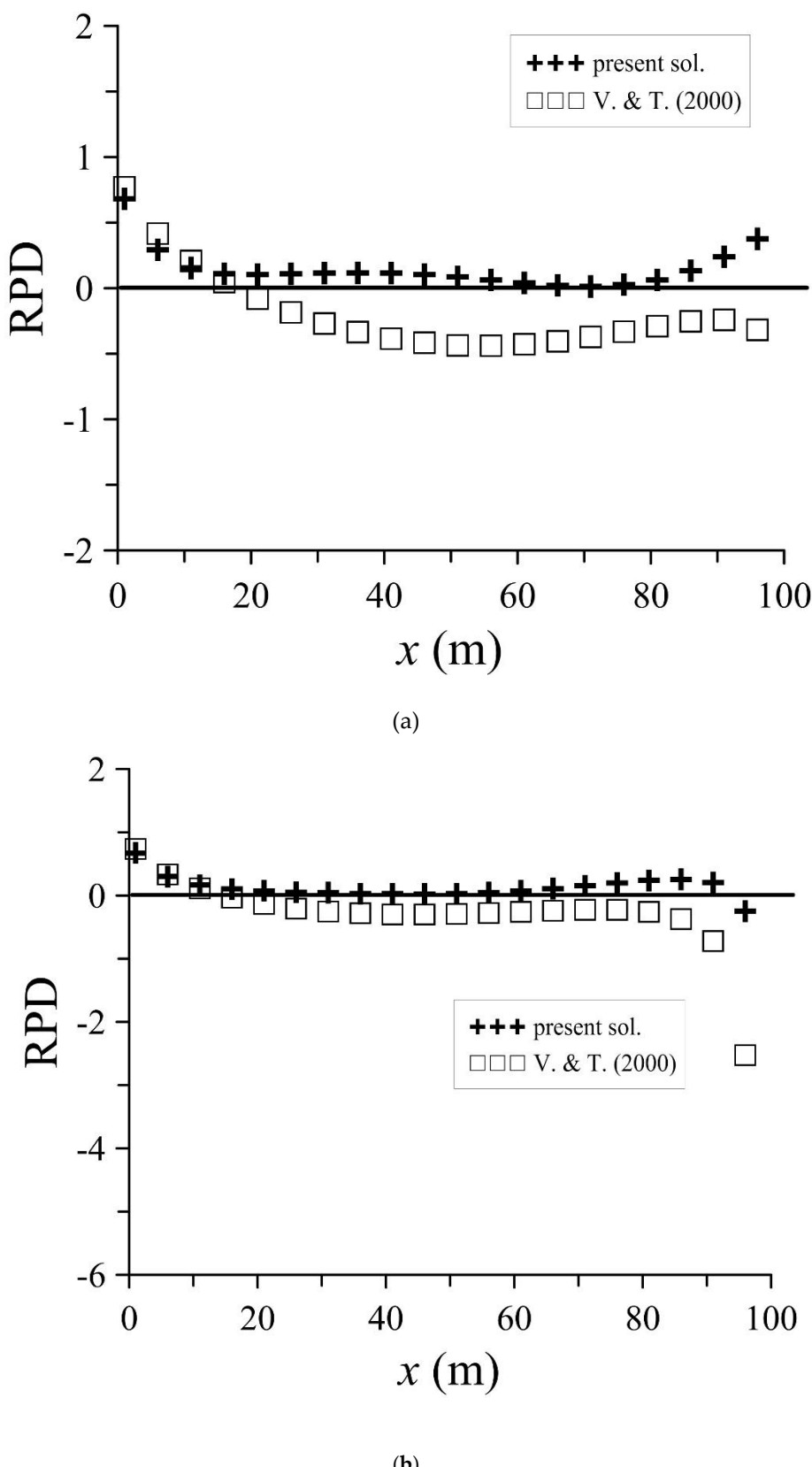

(a)

(b)

**Figure 5.** Relative percentage difference (RPD) between analytical and numerical solutions. (**a**) $\theta = 2°$ (**b**) $\theta = 6°$ ($t = 1$ day).

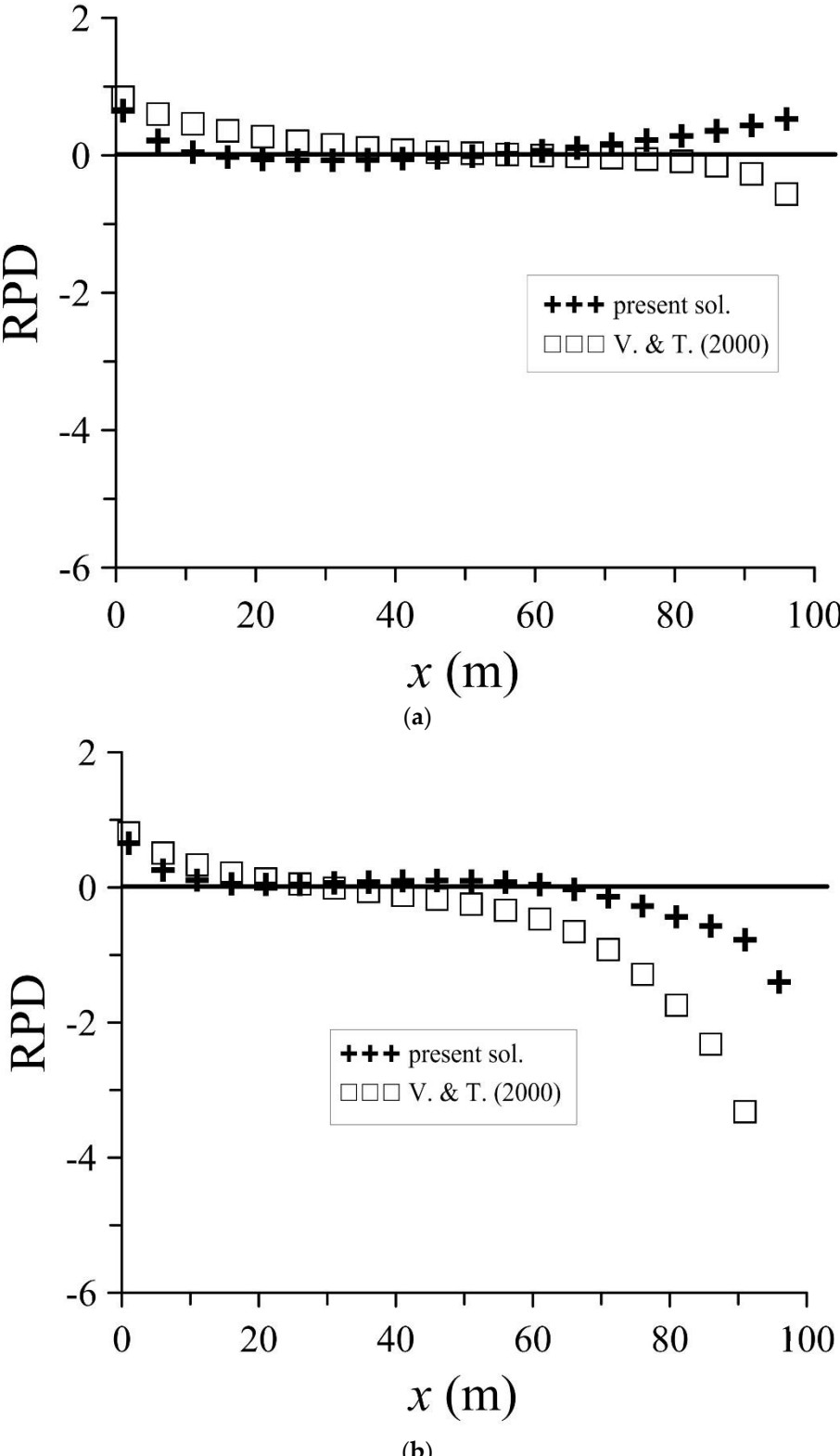

**Figure 6.** RPD between analytical and numerical solutions. (**a**) $\theta = 2°$ (**b**) $\theta = 6°$ ($t = 3$ days).

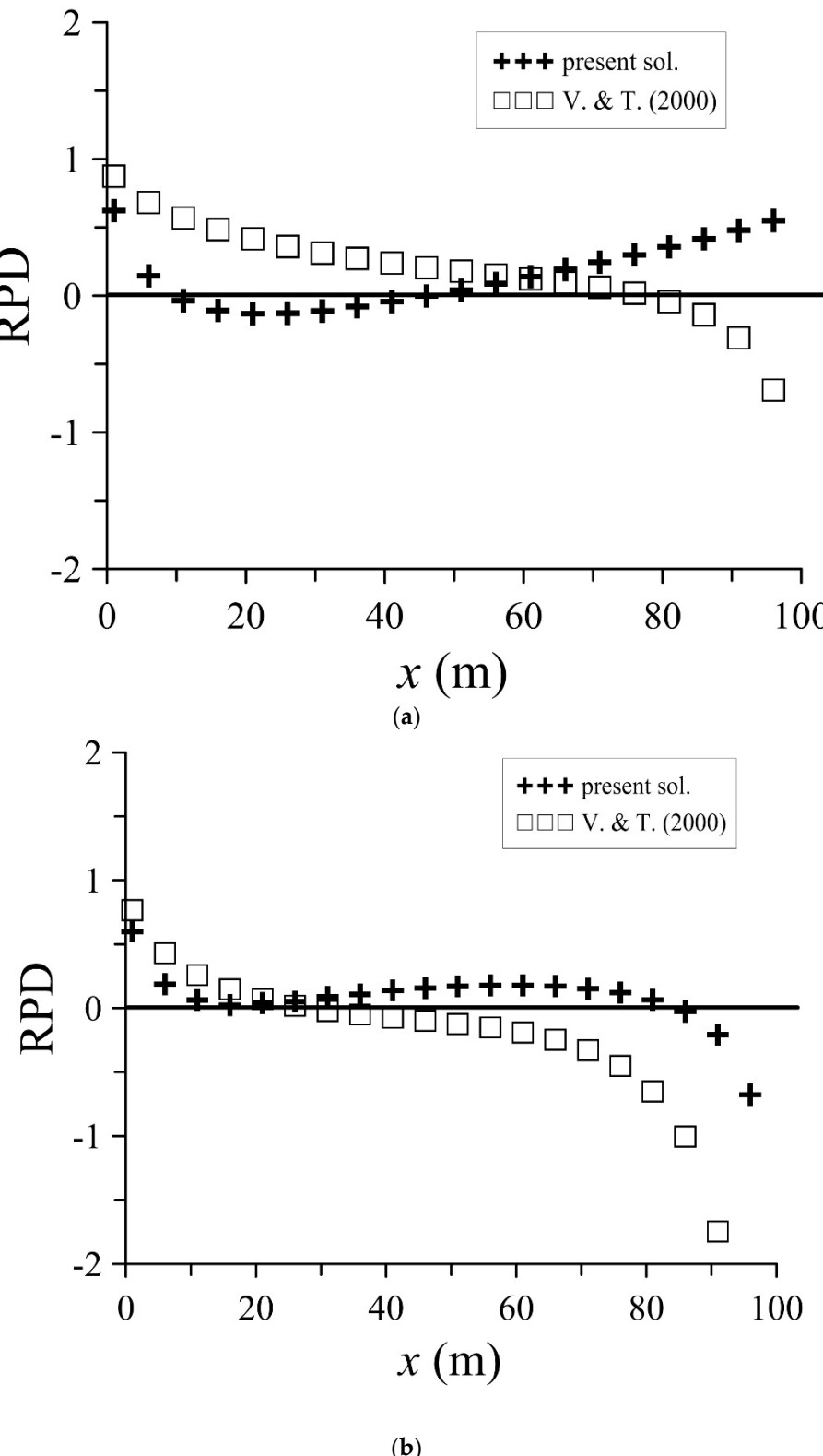

**Figure 7.** RPD between analytical and numerical solutions. (**a**) $\theta = 2°$ (**b**) $\theta = 6°$ ($t = 5$ days).

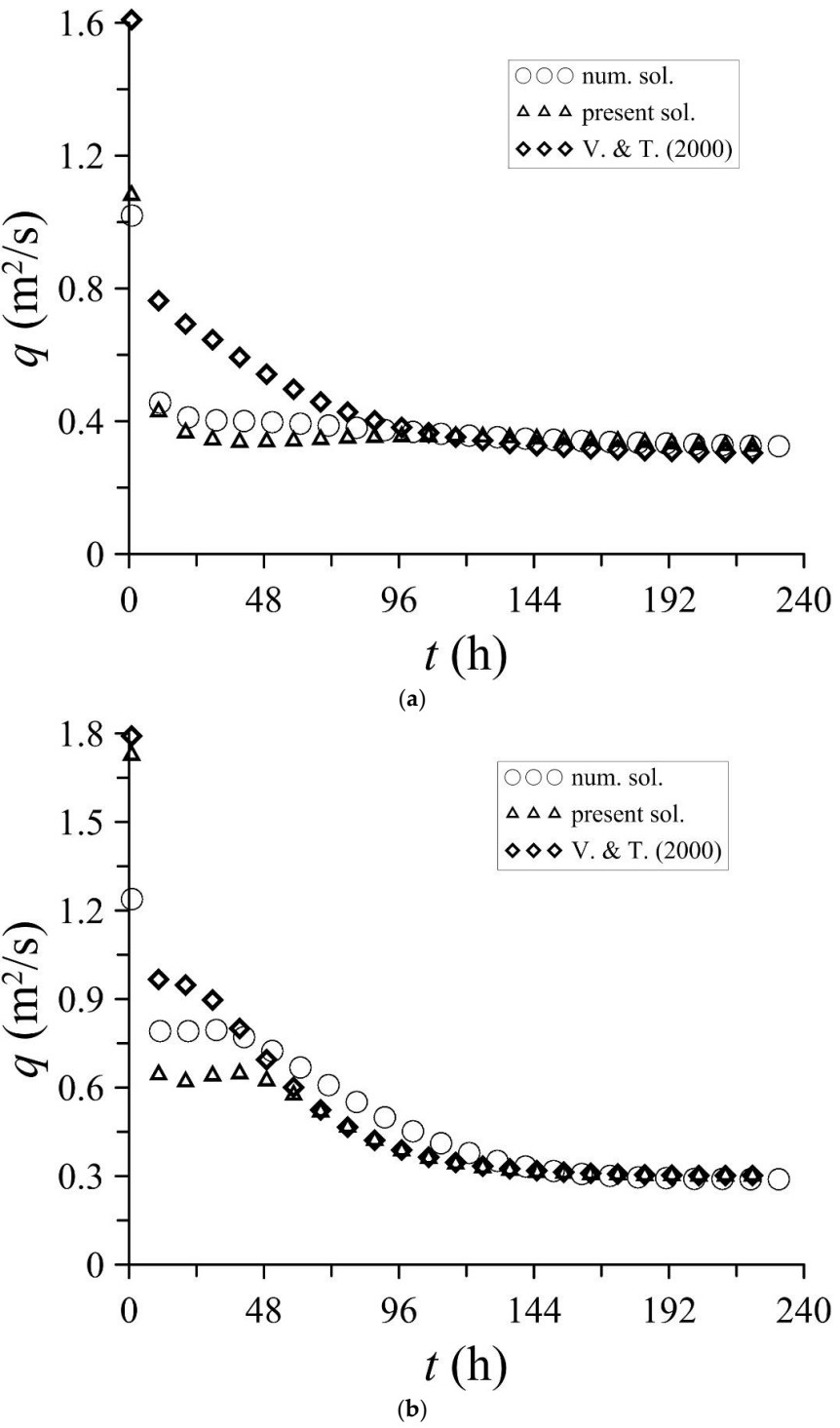

**Figure 8.** Transient variation of outflow under constant recharge for (**a**) $\theta = 2°$ (**b**) $\theta = 6°$.

*3.2. Comparison of Unsteady State and Quasi-Steady State*

Moreover, Verhoest and Troch [7] adopted a quasi-steady state method to calculate the groundwater response of a hillslope for a temporally changing recharge rate. However, the proposed unsteady state solutions could be directly applied to the same hillslope case without requiring any extra treatment, as depicted in Figure 9. Note that the discrepancy between both solutions is not large, and the slight difference primarily arises from the temporal treatment conducted in the study of Verhoest and Troch [7].

In summary, Verhoest and Troch [7] used the Laplace transform method to solve the partial differential specified by Equation (4); however, that inverse Laplace transform is extremely difficult to obtain, even by applying a complex inversion formula for an inverse Laplace transform. Arfken and Weber [8] used a Bromwich integral to overcome the problem; however, convergence could only be approached after a lengthy calculation. When the inclined angle $\theta$ is equal to 2° and 6°, the summation in their solution requires the first 999 terms to reach convergence. However, the proposed solutions obtained by the generalized integral transformation method only require 15 terms to obtain convergence; a reasonable example would be $10^{-5}$ m for the groundwater level and $10^{-3}$ m²/day for the outflow.

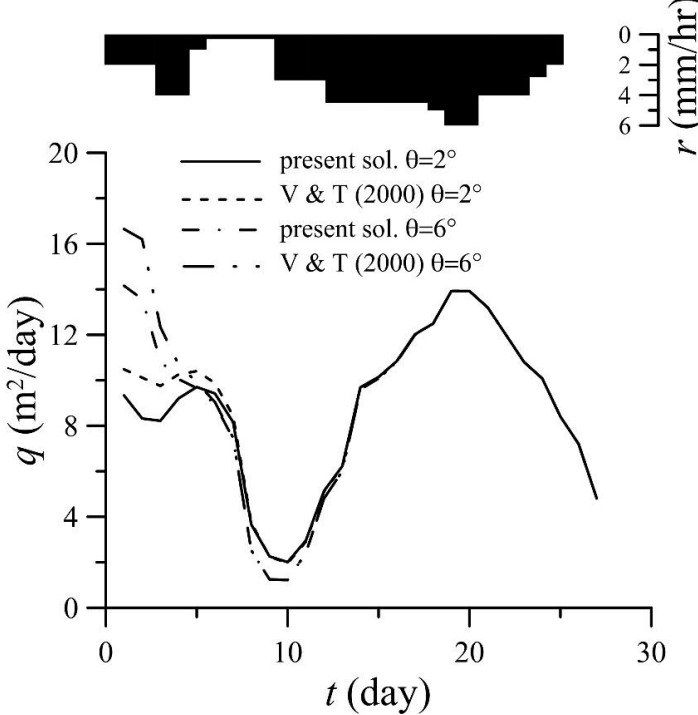

**Figure 9.** Variation in the outflow corresponding to varying recharge rates, as illustrated in the study of Verhoest and Troch [7].

## 4. Conclusions

A generalized integral transformation method can provide an improved solution to a linearized Boussinesq equation for a sloping unconfined aquifer. The presented analytical results combine the effect of the bottom slope and the time-varying recharge pattern on the water table fluctuations. Owing to the limitations and difficulties of directly measuring the groundwater level, we developed a mathematical model such that we can predict or simulate the variation in the groundwater level that can be affected by any rainfall recharge rates. Some conclusions are proposed as follows.

1. According to the error analysis, in the case of a constant recharge rate for a sloping aquifer, the results of the proposed solution are better than the results proposed by Verhoest and Troch [7] after comparing with the numerical solutions; therefore, the present analytical solution appears to be more feasible than that proposed in a previous study.
2. The proposed solutions reach the convergence criteria faster than the solutions of Verhoest and Troch [7], thus saving computation time.
3. The present solution can be directly applied to unsteady recharge rate cases without the requirement of the quasi-steady state method which was employed in the study of Verhoest and Troch [7].

**Author Contributions:** Conceptualization, P.-C.H.; Methodology, P.-C.H.; Software, M.-C.W.; Validation, M.-C.W.; Formal Analysis, M.-C.W.; Investigation, P.-C.H.; Resources, P.-C.H.; Data Curation, M.-C.W.; Writing-Original Draft Preparation, M.-C.W.; Writing-Review & Editing, P.-C.H.; Visualization, P.-C.H.; Supervision, P.-C.H.; Project Administration, P.-C.H.; Funding Acquisition, P.-C.H.

**Funding:** This research was funded by "the Ministry of Science and Technology of Taiwan, grant number: MOST 106-2313-B-005 -007 –MY2:" and "The APC was funded by "the Ministry of Science and Technology of Taiwan".

**Acknowledgments:** This study was financially supported by the Ministry of Science and Technology of Taiwan under Grant No.: MOST 106-2313-B-005 -007 –MY2. In the meanwhile, this manuscript was mostly edited by Wallace Academic Editing.

**Conflicts of Interest:** The authors declare no conflict of interest. The funders had no role in the design of the study; in the collection, analyses, or interpretation of data; in the writing of the manuscript, and in the decision to publish the results.

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
