# Peer review of "Improved Solutions to the Linearized Boussinesq Equation with Temporally Varied Rainfall Recharge for a Sloping Aquifer"

_water, doi:10.3390/w11040826_

Round 1
Reviewer 1 Report
The aim of the article was to present a modified one-dimensional Boussinesq equation for groundwater flow in a sloping aquifer with showing the superiority of their solution relative to previous study. The paper is well organized and the methods and results were clearly presented. I believe the manuscript was well written. I have a few comments, which I believe, can improve the manuscript.
1. The presented analytical solution can consider the effects of the bottom slope and the time-varing recharge pattern on the water table fluctuations. However, the the effect of the time-varing recharge pattern was not sufficiently highlighted relative to the effect of the slope. The most part of the analyses showed the compared results for the case of a constant recharge rate for a sloping aquifer.
2. In the section of 3.1, comparison was made for the conditions of varing slope and constant recharge(3mm/hr). It would be better to show the response according to several recharge.
3. The abrupt changes of water table around x=20, x=65 were shown in Figure 2. Sme disscusion about the reason would be required.
4. It is required to add some description of the reason why the previous V&T solution shows the shifts to the right as indicated in Figure 4.
5. In Figure 6(a), the presented solusion shows some shift to the left.
6. The first sentence of the conclusions may not be impressive.
Author Response
1. Thank you for your comment. The main goal of the present solution is to improve the previous solutions proposed by Verhoest and Troch (2000). In their study, the recharge rate is considered constant. Therefore, Figures 2, 4, 6 are presented to compared with V&T solutions to show the difference under the same parameters and constant recharge rate. Furthermore, the present solution can deal with time-varying recharge problem that V&T could not. Both solutions can handle the problem for a sloping aquifer, so we do not emphasize the sloping effect. However, we add the case of sloping effect under time-varying recharge rate in Figure 9.
2. Thank you for your comment. In order to compare the differences with V&T solution, the recharge rate and the slope are selected according to Verhoest and Troch (2000). In their study, they only show the result for the constant recharge rate 3mm/hr.
3. Thank you for your comment. We improved the drawing by reducing the size of delta(x), and found the curve get smoother. Thanks a lot.
4. Thank you for your comment. Because the numerical solutions consider full nonlinear equation, the numerical solution can be regarded as a criterion. It can be seen from Figure 4 that the difference between the analytical solution of V & T (2000) and the nonlinear numerical solution is more than that of this solution. This shows the present solution is more correct that V&T solution. Firstly, V&T applied the Laplace Transform method to solve the linearized governing equation, and employed the complex inversion formula to find the solution by making use of the Bromwich integral. This process encountered with a convergence problem. (This had been described in the second paragraph of Sec. 3.2) Secondly, V&T solved the linearized governing equation instead of the fully nonlinear one, and their solution got response to the sloping effect slower than the numerical solution. (This is added to Sec. 3.1) I think this might be the reasons of the previous V&T solution shifting to the right in Figure 4.
5. Thank you for your comment. The present solution made response to the sloping effect faster than the nonlinear solution owing to the linearization. This is added to
6. Thank you for your comment. The “alternative” is changed to “improved”.
Reviewer 2 Report
This paper is valuated as an interesting scientific work trying to present a generalized integral transformation method to solve the linearized Boussinesq equation that governs the groundwater level in a sloping unconfined aquifer with an impermeable bottom.
• The specific research work could be appreciated as a notable contribution to developing an analytical solution, which is presented as more workable than that of previous relevant studies.
• The subject is within the topics of the Water Journal.
• The manuscript is clearly written following a structure that contains information, analysis and elaboration results and discussion documented and presented in a quite informative and explanatory way.
My final recommendation is that the manuscript should be accepted in its present form.
Author Response
Thanks for your hard work of reviewing, and thank you very much for your encouragement.